# Risk Factors Predictive of Contralateral Recurrence of Upper Tract Urothelial Carcinoma Include Chronic Kidney Diseases and Postoperative Initiation of Dialysis

**DOI:** 10.3390/cancers17040664

**Published:** 2025-02-16

**Authors:** Yi-Ru Wu, Ching-Chia Li, Yung-Shun Juan, Wei-Ming Li, Wen-Jeng Wu, Tsu-Ming Chien

**Affiliations:** 1Department of Urology, Kaohsiung Medical University Hospital, Kaohsiung 80708, Taiwan; k3329985@gmail.com (Y.-R.W.); ccli1010@hotmail.com (C.-C.L.); juanuro@gmail.com (Y.-S.J.); u8401067@yahoo.com.tw (W.-M.L.); wejewu@kmu.edu.tw (W.-J.W.); 2Department of Urology, Kaohsiung Medical University Gangshan Hospital, Kaohsiung 820111, Taiwan; 3Department of Urology, School of Postbaccalaureate Medicine, College of Medicine, Kaohsiung Medical University, Kaohsiung 80708, Taiwan

**Keywords:** upper tract urothelial carcinoma, contralateral recurrence, chronic kidney diseases, hemodialysis

## Abstract

Upper urinary tract urothelial carcinoma is a relatively uncommon urological cancer. In the past, there has been extensive research on the risk factors for recurrence, with recurrences primarily occurring in the bladder. Patients have a low chance of contralateral recurrence after surgery; however, there is very little research on the follow-up of patients with poor renal function who require dialysis postoperatively. This study focuses on this specific population. We found that patients with poor renal function, especially those who require new initiation of dialysis, have a higher chance of contralateral recurrence of cancer compared to other groups. Additionally, these patients face increased difficulties in postoperative imaging examinations due to their poor renal function. We recommend that, given the high risk, even though examinations are challenging, there is a greater need for proactive monitoring of the possibility of recurrence on the contralateral side.

## 1. Introduction

Upper tract urothelial carcinoma (UTUC) represents a heterogeneous category of urothelial carcinomas that arise within the upper urinary tract, specifically in the renal pelvis and ureter. This condition constitutes roughly 5% of all urothelial carcinomas [1]. While UTUC is generally considered rare in many Western nations, it exhibits epidemic potential in certain localized regions [2]. Raman et al. [3] documented an increase in the overall incidence of UTUC in the United States, with rates increasing from 1.88 to 2.06 cases per 100,000 person/year between 1973 and 2005. Furthermore, an unusually elevated incidence of UTUC has been observed in certain areas of the Balkans and Taiwan [2]. UTUC also exhibits aggressive behavior, with 60% of cases classified as invasive at the time of diagnosis in contrast to 15–25% of cases of urothelial carcinoma of the bladder, resulting in diminished survival rates [4]. Moreover, despite optimal treatment, UTUC is associated with a significantly elevated risk of disease recurrence and progression, particularly in its advanced stages [5].

Chronic kidney disease (CKD) has been identified as an independent risk factor associated with increased incidence rates of UTUC [6]. For patients with poor renal function who undergo surgery for UTUC, there may be irreversible effects on renal function, and they may even need to start receiving dialysis. A collaborative international study has demonstrated that the increased risk of UTUC among patients with chronic kidney disease can primarily be attributed to the influence of pre-existing urologic conditions and heightened vulnerability to viral-induced carcinogenesis [7]. 

The majority of prior research on UTUC has concentrated on the prognostic factors associated with cancer progression, intravesical recurrence, and survival outcomes. Nevertheless, there is a paucity of studies that specifically address the indicators for predicting metachronous contralateral recurrence. The occurrence of contralateral UTUC following the excision of the primary lesion is relatively rare, with an estimated incidence ranging from 0.6% to 6.9% [8,9,10,11,12]. The identification of patients at risk of contralateral recurrence is crucial for facilitating early detection, thereby potentially averting the need for additional radical nephroureterectomy (RNU) procedures for recurrent contralateral tumors and the associated risk of permanent dialysis. 

Some research investigations have sought to examine the risk factors associated with the contralateral recurrence of UTUC [8,9,10,11,12]. Nevertheless, various heterogeneous clinical characteristics remain inadequately defined. To date, there is a limited body of research examining the relationship between dialysis and patients with UTUC following RNU. The aim of the current investigation was to assess the effects of dialysis on patients with UTUC undergoing surgical treatment, as well as to identify predictive factors associated with contralateral recurrence. 

## 2. Materials and Methods

### 2.1. Patients

This study included patients who underwent either open or laparoscopic RNU with bladder cuff excision for non-metastatic UTUC at Kaohsiung Medical University Hospital in Kaohsiung, Taiwan, from 2001 to 2015. This research received approval from the institutional review board (KMUH-IRB-20120138). Participants were categorized into two groups based on contralateral recurrence during the follow-up. The dialysis group consisted of patients diagnosed according to ICD-9-CM codes and procedure codes prior to the management of UTUC at our institution. Clinical parameters, including demographic data, pathological characteristics, oncological follow-up, and causes of mortality, were retrospectively gathered. Patients who received neoadjuvant chemotherapy or radiotherapy, had concurrent bladder tumors, presented with acute hematological disorders, or possessed incomplete clinical records were excluded from this study. Tumor staging was conducted in accordance with the 2002 American Joint Committee on Cancer TNM classification system. All cases were evaluated by two pathologists and subsequently reclassified as low or high grade based on the 2004 World Health Organization grading system. Renal function was assessed utilizing the estimated glomerular filtration rate (eGFR) derived from the creatinine-based formula established by the Chronic Kidney Disease Epidemiology Collaboration (CKD-EPI) [13]. CKD stages 0 and 1 were characterized by normal glomerular filtration rate (eGFR) values.

### 2.2. Follow-Up

Postoperatively, outpatient clinic visits were scheduled at three-month intervals during the initial two years, followed by six-month intervals in the subsequent two years. From the fifth year onward, annual follow-ups were conducted for patients demonstrating no signs of disease. Comprehensive assessments, including detailed medical history, physical examinations, urine cytology, cystoscopy, and serial imaging, were performed in accordance with established surveillance protocols. Tumor recurrence was defined as local recurrence at the surgical site, involvement of regional lymph nodes, or the presence of distant metastases. Tumors that developed in the bladder or the contralateral upper urinary tract were classified as metachronous and were not considered as recurrences of the disease. Patients who underwent bladder tumor excision or cystectomy more than three months following RNU were classified as having experienced bladder recurrence. Adjuvant chemotherapy and radiation therapy were administered to 99 and 44 patients, respectively, based on factors such as pathological stage, performance status, renal function, and patient consent for treatment.

### 2.3. Statistical Analysis 

The differences among categorical variables were assessed using the χ2 test or Fisher’s exact test. The Kaplan–Meier method was employed to evaluate the influence of clinicopathological variables on contralateral recurrence-free survival. The duration of survival was calculated from the date of radical nephroureterectomy (RNU) until the occurrence of contralateral recurrence or the date of the most recent follow-up. The log-rank test was utilized to compare the survival curves.

The Kaplan–Meier method was also applied to estimate the correlation of contralateral recurrence with overall survival (OS), cancer-specific survival (CSS), and bladder recurrence-free survival (BRFS). Survival rates were recorded from the date of RNU until death, the date of achieving cancer-free status, bladder recurrence, or the most recent follow-up visit. The log-rank test was used to compare the survival curves. Only prognostic factors that exhibited statistical significance in the univariate analysis were included in the multivariate Cox proportional hazards model to identify independent predictors for OS, CSS, and BRFS. Additionally, independent risk factors associated with contralateral recurrence were identified. A significance level of *p* < 0.05 was established. All statistical analyses were performed using SPSS version 20.0 (SPSS Inc., Chicago, IL, USA).

## 3. Results

A total of 593 patients underwent radical nephroureterectomy (RNU), among which 42 patients (7.1%) presented with concurrent non-muscle invasive upper urinary tract urothelial carcinoma (UBUC), 6 patients (1.0%) had concurrent muscle invasive UBUC, 5 patients (0.8%) exhibited contralateral UTUC, and 3 patients (0.5%) experienced paraneoplastic leukocytosis. Consequently, 537 patients (contralateral recurrence, 31 (5.7%) patients; non-recurrence 506 (94.2%) patients) were included in the present study.

Preoperative ureterorenoscopic (URS) examination and biopsy were conducted in 456 patients (85.0%), while 54 patients (10.1%) underwent image-guided biopsy, and 27 patients (5%) displayed suspicious imaging characteristics. Among the cohort, 94 patients (17.5%) were on dialysis prior to RNU, and 16 patients (3%) progressed to end-stage renal disease (ESRD), necessitating permanent dialysis postoperatively. 

Table 1 delineates the clinical and pathological profiles of the patients. The mean age of patients who underwent RNU was 67.1 ± 10.4 years, with the contralateral recurrence group averaging 64.1 ± 11.9 years and the non-recurrence group averaging 67.4 ± 10.2 years (*p* = 0.142). The mean follow-up duration following surgery was 42.1 ± 33.5 months, with the contralateral group having a mean follow-up of 36.8 ± 28.8 months and the non-recurrence group 43.4 ± 31.2 months. No significant difference in follow-up duration was noted between the two groups.

Patients with contralateral recurrence exhibited a higher prevalence of female sex, symptomatic hydronephrosis, a history of bladder cancer, advanced CKD, and postoperative hemodialysis compared to those without recurrence, as illustrated in Table 1.

### 3.1. OS, CSS, and BRFS

In our study cohort, a total of 134 patients, representing 24.9%, experienced mortality events, which included 7 cases of contralateral recurrence and 127 cases of non-recurrence. The OS rates at 3 and 5 years were recorded at 76.1% and 67.2%, respectively. Kaplan–Meier analysis revealed no significant association between OS rates and contralateral recurrence status (*p* = 0.243; see Figure 1A). Univariate analysis identified several factors correlated with poorer OS, including age over 65 years (*p* = 0.045), smoking status (*p* = 0.042), a history of bladder cancer (*p* = 0.045), open surgical procedures (*p* = 0.016), tumor location in the ureter (*p* = 0.021), advanced T-stage classification (*p* < 0.001), multifocality of the tumor (*p* < 0.001), lymph node invasion (*p* < 0.001), perineural invasion (*p* < 0.001), high-grade tumor classification (*p* = 0.002), and the administration of adjuvant chemotherapy (*p* < 0.001). Furthermore, smoking (*p* = 0.014), advanced T stage (*p* < 0.001), and adjuvant chemotherapy (*p* < 0.001) were identified as independent risk factors contributing to shorter OS (Table 2).

A total of 84 patients (15.6%) experienced cancer-specific mortality events, which included 6 cases of contralateral recurrence and 78 cases of non-contralateral recurrence. The CSS rates at 3 and 5 years were recorded at 84.6% and 76.9%, respectively. Kaplan–Meier analysis indicated that the CSS rates were not significantly associated with the status of contralateral recurrence (*p* = 0.243; see Figure 1B). Univariate analysis identified several factors correlated with poorer CSS, including advanced stages of CKD (*p* = 0.012), a history of bladder cancer (*p* = 0.037), dialysis status (*p* = 0.003), presence of ureteral tumors (*p* = 0.039), advanced T stage (*p* < 0.001), multifocal tumors (*p* < 0.001), lymph node invasion (*p* < 0.001), perineural invasion (*p* < 0.001), high-grade tumors (*p* < 0.001), and the administration of adjuvant chemotherapy (*p* < 0.001). Furthermore, multivariate analysis confirmed that a history of bladder cancer (*p* = 0.030), pT4 tumor classification (*p* < 0.001), multifocal tumor presence (*p* = 0.037), and the use of adjuvant chemotherapy (*p* < 0.001) were independent risk factors associated with shorter CSS (Table 2).

A cohort of 151 patients, constituting 28.1% of the study population, experienced bladder recurrence. This group included 18 instances of contralateral recurrence and 133 cases of non-recurrence. The overall rates of BRFS at 3 and 5 years were determined to be 77.3% and 63.9%, respectively. Kaplan–Meier analysis indicated no statistically significant relationship between bladder recurrence rates and the status of contralateral recurrence (*p* = 0.178; refer to Figure 1C). Univariate analysis identified several factors associated with a decreased BRFS, including a history of bladder cancer (*p* < 0.001), preoperative hemodialysis (*p* = 0.040), preoperative hydronephrosis (*p* = 0.022), advanced T-stage classification (*p* = 0.010), and the administration of adjuvant chemotherapy (*p* < 0.001). Additionally, a history of bladder cancer (*p* < 0.001), preoperative hydronephrosis (*p* = 0.030), advanced T-stage classification (*p* = 0.042), and adjuvant chemotherapy (*p* < 0.001) were recognized as independent risk factors associated with a shorter BRFS (Table 2).

### 3.2. Contralateral Recurrence 

In our study cohort, 31 patients (5.8%) experienced contralateral recurrence, comprising 25 females and 6 males. The contralateral recurrence-free survival rates at 3 and 5 years were 81.7% and 79.0%, respectively. As illustrated in the preceding figure, contralateral recurrence did not exhibit a correlation with OS, CSS, or BRFS. 

Kaplan–Meier analysis revealed a statistically significant decrease in the contralateral recurrence-free survival rate among patients identified as female (Figure 2A, *p* = 0.040), those with a history of bladder cancer (Figure 2B, *p* < 0.001), presenting with multiple tumors (Figure 2C, *p* = 0.011), patients with advanced CKD (Figure 2D, *p* < 0.001), and those requiring postoperative dialysis (Figure 2E, *p* < 0.001). Conversely, preoperative hemodialysis status did not demonstrate a significant association with contralateral recurrence (Figure 2F, *p* =0.08). Univariate analysis further indicated that a history of bladder cancer (*p* < 0.001), the presence of multiple tumors (*p* = 0.020), advanced CKD (*p* < 0.001), and the initiation of new dialysis (*p* = 0.001) were correlated with poorer contralateral recurrence-free survival outcomes. In the multivariate analysis, a history of bladder cancer (hazard ratio (HR), 3.19; 95% confidence interval (CI), 1.2–8.4; *p* = 0.018), the requirement for new hemodialysis postoperatively (HR, 5.34; 95% CI, 1.3-25.6; *p* = 0.034) and advanced CKD (HR, 2.52; 95% CI, 1.4–4.9; *p* = 0.021) were identified as independent risk factors contributing to an increased rate of contralateral recurrence (Table 3).

## 4. Discussion

To the best of our knowledge, this study represents the inaugural investigation into the significance of postoperative initiation of hemodialysis concerning the contralateral recurrence of UTUC following unilateral RNU. However, we observed no statistically significant differences in OS, CSS, or BRFS rates between patients with contralateral recurrence and those without. Our findings corroborate previous research that identified advanced CKD [9,11,12] and a history of bladder cancer [8,9,10,12] as independent risk factors for contralateral recurrence.

Patients diagnosed with CKD exhibit an elevated likelihood of developing incident urothelial carcinoma [6] and experience a higher mortality rate specifically associated with urothelial cancer [14]. Furthermore, CKD is identified as a significant risk factor for UTUC [9,11,12] and is correlated with the disease’s aggressiveness [15]. Conversely, individuals with CKD do not demonstrate an increased risk for prostate, colorectal, lung, or breast cancers [6]. Consistent with prior research [9,11,12], our findings suggest that advanced CKD serves as an independent predictor for the recurrence of contralateral UTUC following RNU. Diminished renal function results in the accumulation of uremic toxins, which are believed to induce oxidative stress and foster a persistent inflammatory milieu, thereby triggering the activation of the immune response [16,17]. Consequently, there is an expansion of circulating proinflammatory lymphoid (CD28null T cells) and myeloid (CD16+ monocytes) cell populations, while the overall efficacy of immune function is compromised [17]. As a result, the cellular immunity of individuals with CKD is significantly impaired. This acquired immunodeficiency may contribute to the development of UTUC and the subsequent metachronous recurrence on the contralateral side.

The prior or concurrent presence of bladder cancer has been associated with an increased risk of contralateral recurrence [8,9,10,12]. This correlation lends partial support to the hypothesis of intraluminal seeding as a potential mechanism underlying the emergence of metachronous contralateral upper urinary tract malignancies. However, due to the infrequency of contralateral retrograde procedures or vesicoureteral reflux, earlier investigations have posited that field cancerization—a theory that explains the occurrence of multiple genetically distinct tumors induced by carcinogenic agents—may significantly contribute to the development of contralateral tumors [11,18]. Our findings indicate that the presence of multiple tumors in cases of contralateral recurrence provides some validation for this hypothesis. In univariate analysis, the presence of multiple tumors was identified as a negative prognostic factor for contralateral recurrence; however, this association did not achieve statistical significance in the multivariate analysis.

Patients diagnosed with ESRD who are undergoing dialysis have been identified as having an elevated risk for various types of cancer [7,19,20]. A population-based study conducted in the United States revealed that the risk of cancer was particularly pronounced for malignancies of the urinary tract [19]. Additionally, another population-based cohort study in Taiwan observed a trend indicating an increased cancer risk among younger ESRD patients, particularly within the first year of initiating dialysis [20]. Patients undergoing dialysis are currently considered to be in a state of immune deficiency in many studies. The first few years after starting dialysis are also the period when cancer is most likely to occur, primarily because the immune system of patients may be affected during this time. Dialysis is a treatment method used to replace kidney function, but this process can lead to immune system suppression, making patients more susceptible to infections and cancer. Additionally, dialysis patients often require long-term medication, which may have side effects that increase the risk of cancer. This may partially explain why newly dialyzed patients in our population are more prone to contralateral recurrence.

Our findings indicate that preoperative hemodialysis does not exhibit a correlation with contralateral recurrence of UTUC. Interestingly, patients who underwent hemodialysis postoperatively displayed a higher rate of contralateral recurrence. Furthermore, subgroup analysis revealed that patients initiating dialysis after surgery experienced an exceptionally elevated recurrence rate. Patients on dialysis experience diminished urine flow, which results in a high concentration of toxic waste products accumulating in the urinary tract and bladder. This prolonged exposure to toxins may elucidate the observed higher incidence of UTUC compared to urinary UBUC in this patient population [21]. 

Smoking is a known adverse factor for UTUC [22]. In our previous studies, we found that the smoking population among men with UTUC indeed has a significant impact on OS. However, in Taiwan, the composition of UTUC shows a slightly higher prevalence in women than in men, and the proportion of women with ESRD and undergoing dialysis is quite high. Due to these factors, we obtained a result in our preliminary analysis indicating that smoking has a marginal statistically significant effect (*p* = 0.074). Previous articles have also shown the adverse effects of smoking and the benefits of quitting. Therefore, we strongly recommend that patients quit smoking as early as possible in order to avoid increasing the risk of recurrence and mortality.

Our prior research identified two significant gender-related disparities: a higher prevalence of CKD and dialysis among women, and an elevated smoking rate among men [23]. We established that the female sex does not serve as an unfavorable prognostic factor for UTUC. Given the increased incidence of CKD in women diagnosed with UTUC, we posited that CKD status might also impact our earlier findings. Few studies have addressed the differences in CKD prevalence between sexes. Our previous study indicated that, after controlling for CKD status, women exhibited more favorable outcomes regarding metastasis. Further exploration is necessary to identify and investigate additional factors contributing to the distinct clinical presentations observed in female patients in Taiwan. Theoretical frameworks explaining the variations in incidence, severity, and prognosis of UTUC between sexes remain undeveloped. Disparities in carcinogenic exposures, routes of entry, or the enzymatic metabolism of environmental agents may elucidate the observed clinical differences. In the current study, Kaplan–Meier analysis revealed a higher rate of contralateral recurrence in women; however, multivariate analysis, after adjusting for other confounding variables, did not demonstrate a significant sex difference in contralateral recurrence rates.

This research is subject to several limitations. Firstly, it was conducted as a retrospective study at a single center. Secondly, the patients included in this study were treated by various surgeons over a span of 14 years. Given the distinct characteristics of Taiwanese patients with UTUC, the generalizability of our results to other populations may be limited. Therefore, further large-scale studies are necessary to validate our findings. The issue of CKD as a potential risk factor for contralateral recurrence is multifaceted. The limitations of imaging techniques in preoperative assessments may hinder the accurate differentiation between synchronous bilateral tumors and de novo tumors. In the context of a retrospective study such as the one presented, it is important to note that not all patients with advanced CKD may have undergone an optimal evaluation of the contralateral kidney prior to surgery. Lastly, the incidence of contralateral recurrence after surgical treatment for UTUC is relatively low. In clinical practice, when managing patients with poor renal function, the risk of postoperative dialysis is also one of our important considerations. This significant prognostic risk factor is currently not given much attention. Although our study has a limited number of cases and may not provide absolute conclusions, we hope that oncologic doctors can pay more attention to this population. Additionally, since postoperative contralateral recurrence is relatively difficult to detect, and new dialysis patients often are not suitable for imaging examinations using contrast agents, we hope to remind everyone to be more aware of the potential risks based on our observed results.

## 5. Conclusions

In conclusion, advanced CKD, a history of bladder cancer, and the initiation of new dialysis postoperatively were identified as independent prognostic indicators for contralateral recurrence in patients with initial unilateral UTUC undergoing RNU. It is advisable for patients exhibiting these three unfavorable characteristics to undergo vigilant surveillance of the contralateral upper urinary tract throughout the follow-up period.

## Figures and Tables

**Figure 1 cancers-17-00664-f001:**
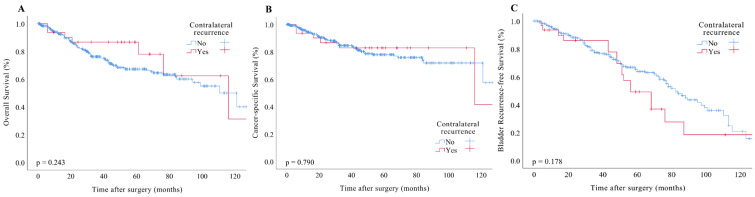
Kaplan-Meier estimates of survival in patients with UTUC based on contralateral recurrence. (**A**), OS. (**B**), CSS. (**C**), BRFS.

**Figure 2 cancers-17-00664-f002:**
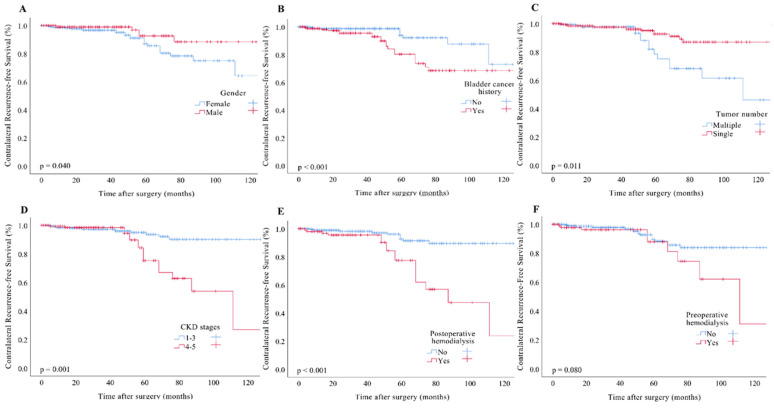
Kaplan–Meier estimates of contralateral survival in patients with UTUC based on (**A**), gender. (**B**), bladder cancer history. (**C**), tumor number. (**D**), CKD status. (**E**), postoperative hemodialysis. (**F**), preoperative hemodialysis.

**Table 1 cancers-17-00664-t001:** Demographics and clinicopathologic characteristics of 537 patients with UTUC according to contralateral recurrence or not.

Variables	Total	Contralateral Recurrence		
		Yes	No	*p* Value
	n = 537	n = 31	n = 506	
Age (Years)				
Mean ± SD	67.1 ± 10.4	64.1 ± 11.9	67.4 ± 10.2	0.142
Over 65 years	336 (62.6)	19 (61.3)	317 (62.6)	0.879
Female gender				** *0.012* **
Yes	317 (59.0)	25 (80.6)	292 (57.7)	
Smoking				0.074
Yes	112 (22.7)	3 (9.7)	119 (23.5)	
BMI (kg/m^2^)				0.790
Over 27	324 (60.3)	18 (58.1)	306 (60.5)	
Preoperative hydronephrosis				** *<0.001* **
Yes	92 (17.1)	13 (41.9)	79 (15.6)	
Bladder cancer history				** *<0.001* **
Yes	170 (31.7)	21 (67.7)	149 (29.4)	
Advanced CKD				** *0.006* **
Yes	159 (29.6)	16 (51.6)	143 (28.3)	
Preoperative HD				0.082
Yes	94 (17.5)	9 (29.0)	85 (16.8)	
Postoperative HD				** *<0.001* **
Yes	106 (19.7)	16 (51.6)	90 (17.8)	
Type of operation				0.328
Open	322 (60.0)	16 (51.6)	306 (60.5)	
Laparoscopic	215 (40.0)	15 (48.4)	200 (39.5)	
Tumor location				0.447
Pyelocaliceal	216 (40.2)	10 (32.3)	206 (40.7)	
Ureteral	224 (41.7)	13 (41.9)	211 (41.7)	
Both	97 (18.1)	8 (25.8)	89 (17.6)	
Multifocality				0.297
Single	374 (69.6)	19 (61.3)	355 (70.2)	
Multifocal	163 (30.4)	12 (38.7)	151 (29.8)	
Pathological tumor stage				0.092
pTis/pTa/pT1	216 (40.2)	19 (61.2)	197 (38.9)	
pT2	150 (27.9)	3 (9.7)	147 (29.1)	
pT3	145 (27.0)	7 (22.6)	138 (27.3)	
pT4	26 (4.8)	2 (6.5)	24 (4.7)	
Grade				0.553
High	405 (75.4)	22 (71.0)	383 (75.7)	
Low	132 (24.6)	9 (29.0)	123 (24.3)	
Adjuvant chemotherapy				0.733
Yes	99 (18.4)	5 (16.1)	94 (18.6)	
Radiation therapy				0.325
Yes	44 (8.2)	4 (12.9)	40 (7.9)	

BMI, body mass index; CKD, chronic kidney disease; ECOG, Eastern Cooperative Oncology Group; UTUC, upper tract urothelial carcinoma; Bold values are statistically significant at *p* < 0.05.

**Table 2 cancers-17-00664-t002:** Univariate and multivariate analyses predicting OS, CSS, and BRFS in patients (n = 537) with UTUC after RNU.

Parameters	OS				CSS				BRFS			
	Univariate Analysis	*p*	Multivariate Analysis	*p*	Univariate Analysis	*p*	Multivariate Analysis	*p*	Univariate Analysis	*p*	Multivariate Analysis	*p*
	HR (95% CI)		HR (95% CI)		HR (95% CI)		HR (95% CI)		HR (95% CI)		HR (95% CI)	
Gender												
Male vs. Female	0.9 (0.6–1.3)	0.447			1.0 (0.6–1.7)	0.927			1.1 (0.6–2.2)	0.788		
Age (Years)												
Over 65 years	1.5 (1.1–2.4)	** *0.045* **	1.4 (0.9–2.3)	0.121	1.5 (0.9–2.5)	0.100			1.0 (0.7–1.3)	0.826		
Smoking												
Yes vs No	1.9 (1.1–3.2)	** *0.022* **	2.1 (1.2–3.9)	** *0.014* **	1.2 (0.8–2.0)	0.376			1.0 (0.7–1.5)	0.836		
BMI (kg/m^2^)												
Over 27 kg/m^2^	1.3 (0.9–2.0)	0.155			1.2 (0.9–1.5)	0.101			1.1 (0.8–1.2)	0.921		
ECOG												
2, 3 vs. 0, 1	1.4 (0.8–2.8)	0.341			1.1 (0.9–2.0)	0.558			1.2 (0.8–2.1)	0.513		
CKD stage												
Stage 2 vs. Stage 0, 1	1.0 (0.4–3.2)	0.890			1.1 (0.4–3.6)	0.511			1.2 (0.4–3.5)	0.5111		
Stage 3 vs. Stage 0, 1	2.4 (0.8–7.4)	0.124			1.2 (0.6–3.4)	0.482			1.2 (0.7–3.5)	0.431		
Stage 4 vs. Stage 0, 1	1.5 (0.3–7.1)	0.735			1.3 (0.5–4.9)	0.823			1.4 (0.2–4.1)	0.865		
Stage 5 vs. Stage 0, 1	1.2 (0.3–3.2)	0.781			1.5 (0.8–4.3)	0.514			1.6 (0.6–4.5)	0.223		
Advanced CKD (stage 4, 5)												
Yes vs. No	1.1 (0.8–1.5)	0.532			1.1 (0.8–1.6)	0.590			1.2 (0.9–1.3)	0.455		
Bladder cancer history												
Yes vs. No	1.5 (1.1–2.3)	** *0.045* **	1.5 (0.9–2.5)	0.066	1.6 (0.9–2.5)	0.071			4.3 (2.2–12.3)	** *<0.001* **	2.8 (2.3–5.6)	** *<0.001* **
Dialysis												
Yes vs. No	1.1 (0.7–1.9)	0.628			1.0 (0.5–1.8)	0.897			2.2 (1.2–3.1)	** *0.040* **	1.2 (0.8–3.1)	0.665
Preoperative hydronephrosis												
Yes vs. No	1.3 (0.8–2.1)	0.375			1.1 (0.6–2.0)	0.777			2.3 (1.1–6.3)	** *0.022* **	1.8 (1.1–5.2)	** *0.030* **
Type of operation												
Laparoscopic vs. open	0.8 (0.7–0.9)	** *0.016* **	0.7 (0.5–1.2)	0.189	0.6 (0.3–0.8)	** *0.007* **	0.7 (0.3–1.0)	0.069	0.7 (0.5–2.2)	0.154		
Tumor location												
Ureteral vs. Pyelocaliceal	1.4 (1.2–2.7)	** *0.021* **	1.1 (0.8–1.5)	0.490	1.0 (0.7–1.7)	0.106			1.2 (0.6–2.3)	0.552		
Both vs. Ureteral	1.3 (0.7–2.4)	0.783			1.3 (0.9–2.7)	0.438			1.4 (0.9–2.2)	0.431		
Both vs. Pyelocaliceal	1.2 (0.8–2.9)	0.342			1.6 (0.7–2.6)	0.304			1.5 (0.8–2.5)	0.212		
Pathologic T stage												
pT2 vs. pTa/pTis/pT1	1.6 (0.6–4.7)	0.389	1.5 (0.6–4.5)	0.693	1.1 (0.6–1.8)	0.914			1.1 (0.6–1.7)	0.922		
pT3 vs. pTa/pTis/pT1	6.2 (4.3–11.6)	** *<0.001* **	3.3 (2.5–8.6)	** *<0.001* **	4.4 (1.7–6.9)	** *<0.001* **	1.7 (1.2–2.4)	** *0.006* **	1.3 (0.7–2.2)	0.882		
pT4 vs. pTa/pTis/pT1	7.2 (6.8–17.4)	** *<0.001* **	2.6 (2.3–9.5)	** *<0.001* **	3.9 (1.1–6.4)	** *<0.001* **	1.4 (0.9–3.2)	0.542	3.1 (1.2–6.5)	** *0.010* **	2.1 (1.2–3.1)	** *0.042* **
Multifocality												
Multifocal vs. Single	4.1 (2.1–7.9)	** *<0.001* **	1.8 (0.8–4.3)	0.163	6.3 (3.2–12.6)	** *<0.001* **	2.4 (1.3–5.9)	** *0.049* **	1.7 (1.3–2.5)	** *0.029* **	1.1 (0.9–2.2)	0.307
Lymph node invasion												
Yes vs. No	5.8 (2.8–12.2)	** *<0.001* **	2.2 (0.9–5.2)	0.069	6.6 (3.2–13.6)	** *<0.001* **	3.6 (1.6–8.0)	** *0.002* **	2.0 (0.8–5.2)	0.110		
Perineural invasion												
Yes vs. No	2.2 (1.4–3.6)	** *<0.001* **	1.1 (0.6–1.9)	0.887	3.4 (2.0–5.7)	** *<0.001* **	1.8 (0.9–3.4)	0.063	1.8 (0.9–4.9)	0.061		
Grade												
High vs. Low	2.3 (1.4–4.1)	** *0.002* **	1.3 (0.8–2.6)	0.394	3.4 (1.6–7.3)	** *0.001* **	2.2 (1.1–4.9)	** *0.046* **	1.1 (0.8–1.2)	0.424		
Adjuvant chemotherapy												
Yes vs. No	3.8 (2.4–6.0)	** *<0.001* **	2.7 (1.6–4.7)	** *<0.001* **	5.7 (3.4–9.5)	** *<0.001* **	3.4 (1.9–6.1)	** *<0.001* **	2.3 (1.2–5.1)	** *<0.001* **	1.9 (1.1–3.8)	** *<0.001* **
Anemia												
Yes vs. No	0.9 (0.6–1.4)	0.792			1.5 (0.9–2.5)	0.143			1.2 (0.8–1.9)	0.122		

BMI, body mass index; CKD, chronic kidney disease; ECOG, Eastern Cooperative Oncology Group; UTUC, upper tract urothelial carcinoma. Bold values are statistically significant at *p* < 0.05.

**Table 3 cancers-17-00664-t003:** Independent risk factors for contralateral recurrence.

Variables	Odds Ratio	95% CI	*p* Value
Bladder cancer history	3.19	1.2–8.4	** *0.018* **
Postoperative new hemodialysis	5.34	1.3–25.6	** *0.034* **
Advanced CKD	2.52	1.4–4.9	** *0.021* **

Bold values are statistically significant at *p* < 0.05.

## Data Availability

The data presented in this study are available upon reasonable request from the corresponding author. The data are not publicly available due to patient privacy.

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
