# Peer review of "Risk Factors Predictive of Contralateral Recurrence of Upper Tract Urothelial Carcinoma Include Chronic Kidney Diseases and Postoperative Initiation of Dialysis"

_cancers, 2025, doi:10.3390/cancers17040664_

Round 1

Reviewer 1 Report

Comments and Suggestions for Authors

The study by Yi Ru focusses on analyzing the effect of dialysis on UTUC patients that are undergoing surgical treatment and identifying the predictive factors related to contralateral recurrence. The manuscript is quite elaborative. Methods and results are well written. I have the following comments:

# Smoking being one of the significant risk factors of UTUC should be written in little detailed manner in results, though it being insignificant in Table 1 (p=0.074).

# Why hasn't author categorized gender as male and female rather written female gender only in Table 1?

Reviewer 2 Report

Comments and Suggestions for Authors

The authors conducted a retrospective study to identify risk factors for metachronous contralateral recurrence of upper urinary tract urothelial cancer (UC) following nephroureterectomy. The study included 593 patients, with a recurrence rate of 5.8%. They found that advanced chronic kidney disease (CKD), the need for hemodialysis (HD) initiation postoperatively, and a history of bladder cancer were independent risk factors for recurrence, as determined by multivariate analysis.

This well-structured manuscript addresses previously underexplored aspects of urothelial cancer. Although upper urinary tract UC occurs less frequently than bladder cancer, it tends to be more malignant and is more prone to progression and recurrence compared to bladder UC. The conclusion is clear, and the topic holds significant clinical relevance for both urologists and oncologists. The findings could influence treatment strategies for this patient population.

The manuscript is well-written and includes two figures and three tables. It cites 21 references, which is appropriate for the scope of this study.

The manuscript can be accepted for publication in its current form.

Author Response

Responses to Reviewer Comments

Reviewer 2

Comments to the Author

The manuscript can be accepted for publication in its current form.

Response : Thank you to the reviewers for their recognition of our work. We will continue to strive to share our results with everyone. Thank you once again.

We hope that we have answered all questions raised by the reviewers and have described the study limitations accurately and adequately. We appreciate very much all of your comments.

Sincerely,

Tsu-Ming Chien, M.D., Ph.D

Department of Urology, Kaohsiung Medical University Hospital, Kaohsiung, Taiwan

Telephone number: +886-7- 3208212

Fax number: +886-7- 3211033

E-mail address: slaochain@gmail.com

Reviewer 3 Report

Comments and Suggestions for Authors

In the present work the authors aim to identify factors predictive of recurrence in the contralateral kidney after radical nephroureterectomy because UTUC. Although retrospective, the series have a good sample size. Unfortunately, there has been quite a number of papers describing risk factors for overall recurrence or metastatic progression after RNU while the contralateral recurrence is very low and consequently to identify a risk factor for becomes challenging.   

While the concept is interesting the project faces several limitations among them the low prevalence of the outcome they aim to predict. This is further stressed in their data with a contralateral UTUC prevalence of 5.7%.

The authors concluded that the need for postoperative dialysis represents a risk factor for contralateral recurrence but then the variable is badly depicted  in table 1 where postoperative dialysis is depictive as a cumulative variable ( pre + post dialysis); as such the figures in postoperative are confounded with the ones in the preoperative  and when the small number of patients that require postoperative dialysis is added to the number  of patients presenting contralateral recurrence the difference in contralateral recurrence becomes significant.  Based on the lack of correction for this confounder, the authors cannot conclude that the need for postoperative dialysis is a risk factor for contralateral recurrence.

Specific comments

In method and results, when looking at the profusion of factors tested in the univariate as potential predictors there is an apparent lack of rationale of some of them, e.g. why to include the type of operation is a risk factor?

The number of patients necessitating dialysis after RNU is only 3% in this series, while a major part of the patients necessitating dialysis were already under it before RNU.  The prevalence of this variable is too small to end up in a sound statistically for a clinically relevant risk factor.   

Time to contralateral recurrence is not described. While mean follow after surgery overall and for the 2 groups (recurrent or not recurrent) the strict time at which the contralateral recurrence appear is not reported.

Table 2 should be clarified, in the variable CKD the reference is Stage 0.1, could please clarify stage CKD stage 0.1?

Table 3 presents the results of the multivariate analysis OR for bladder cancer history, postoperative haemodialysis and advanced CKD are high and clinically significant, however the wider 95% CI shown for all the independent factors means that the strength of the association between exposure and event is low as it the level of precision and thus the clinical relevance.  

The problem when considering the advanced CKD as a risk factor for contralateral recurrence is complex. As imaging may not be as accurate as needed in the preoperative evaluation, it is difficult to determine if the contralateral tumour was synchronic bilateral or “de novo” tumour.   In a retrospective series as the present not all the patients with advanced CKD preoperatively may have had and optimal evaluation of the contralateral kidney.  This difficulty should be stressed in the discussion.

Author Response

Please refer to the attached file, thank you.

Reviewer 4 Report

Comments and Suggestions for Authors

This is a well written manuscript presenting on UTUC issues from a different angle. The data are well analyzed and presented. I have no specific comments to this publication. The authors deserve to be complimented with their work

Author Response

Responses to Reviewer Comments

Reviewer 4

Comments to the Author

This is a well written manuscript presenting on UTUC issues from a different angle. The data are well analyzed and presented. I have no specific comments to this publication. The authors deserve to be complimented with their work

Response : Thank you to the reviewers for their recognition of our work. We will continue to strive to share our results with everyone. Thank you once again.

We hope that we have answered all questions raised by the reviewers and have described the study limitations accurately and adequately. We appreciate very much all of your comments.

Sincerely,

Tsu-Ming Chien, M.D., Ph.D

Department of Urology, Kaohsiung Medical University Hospital, Kaohsiung, Taiwan

Telephone number: +886-7- 3208212

Fax number: +886-7- 3211033

E-mail address: slaochain@gmail.com

Reviewer 5 Report

Comments and Suggestions for Authors

This is a valuable contribution to the literature and the results shed light on common clinical questions that arise in conversation with patients with upper tract urothelial cancer. The results are also immediately actionable and potentially very impactful. Further prospective validation would be of value. The study is scientifically sound and reads well.

Author Response

Responses to Reviewer Comments

Reviewer 5

Comments to the Author

This is a valuable contribution to the literature and the results shed light on common clinical questions that arise in conversation with patients with upper tract urothelial cancer. The results are also immediately actionable and potentially very impactful. Further prospective validation would be of value. The study is scientifically sound and reads well.

Response : Thank you to the reviewers for their recognition of our work. We will continue to strive to share our results with everyone. Thank you once again.

We hope that we have answered all questions raised by the reviewers and have described the study limitations accurately and adequately. We appreciate very much all of your comments.

Sincerely,

Tsu-Ming Chien, M.D., Ph.D

Department of Urology, Kaohsiung Medical University Hospital, Kaohsiung, Taiwan

Telephone number: +886-7- 3208212

Fax number: +886-7- 3211033

E-mail address: slaochain@gmail.com